# Universal Quantum Computing and Three-Manifolds

**Michel Planat** [1,*] , **Raymond Aschheim** [2] , **Marcelo M. Amaral** [2] **and Klee Irwin** [2]

[1] Institut FEMTO-ST CNRS UMR 6174, Université de Bourgogne/Franche-Comté, 15 B Avenue des Montboucons, F-25044 Besançon, France

[2] Quantum Gravity Research, Los Angeles, CA 90290, USA; raymond@QuantumGravityResearch.org (R.A.); Marcelo@quantumgravityresearch.org (M.M.A.); Klee@quantumgravityresearch.org (K.I.)

* Correspondence: michel.planat@femto-st.fr

**Abstract:** A single qubit may be represented on the Bloch sphere or similarly on the 3-sphere $S^3$. Our goal is to dress this correspondence by converting the language of universal quantum computing (UQC) to that of 3-manifolds. A magic state and the Pauli group acting on it define a model of UQC as a positive operator-valued measure (POVM) that one recognizes to be a 3-manifold $M^3$. More precisely, the $d$-dimensional POVMs defined from subgroups of finite index of the modular group $PSL(2, \mathbb{Z})$ correspond to $d$-fold $M^3$- coverings over the trefoil knot. In this paper, we also investigate quantum information on a few 'universal' knots and links such as the figure-of-eight knot, the Whitehead link and Borromean rings, making use of the catalog of platonic manifolds available on the software SnapPy. Further connections between POVMs based UQC and $M^3$'s obtained from Dehn fillings are explored.

**Keywords:** quantum computation; IC-POVMs; knot theory; three-manifolds; branch coverings; Dehn surgeries

**PACS:** 03.67.Lx; 03.65.Wj; 03.65.Aa; 02.20.-a; 02.10.Kn; 02.40.Pc; 02.40.Sf

**MSC:** 81P68; 81P50; 57M25; 57R65; 14H30; 20E05; 57M12

---

*Manifolds are around us in many guises.*

*As observers in a three-dimensional world, we are most familiar with two-manifolds: the surface of a ball or a doughnut or a pretzel, the surface of a house or a tree or a volleyball net...*

*Three-manifolds may be harder to understand at first. But as actors and movers in a three-dimensional world, we can learn to imagine them as alternate universes.*

(William Thurston [1]).

## 1. Introduction

Mathematical concepts pave the way for improvements in technology. As far as topological quantum computation is concerned, non-abelian anyons have been proposed as an attractive (fault-tolerant) alternative to standard quantum computing which is based on a universal set of quantum gates [2–5]. Anyons are two-dimensional quasiparticles with world lines forming braids in space-time. Whether non-abelian anyons do exist in the real world and/or would be easy to create artificially, is still open to discussion. In this paper, we propose an alternative to anyon-based universal quantum computation (UQC) thanks to three-dimensional topology. Our proposal relies on appropriate 3-manifolds whose fundamental group is used for building the magic states for UQC. Three-dimensional topological quantum computing would federate the foundations of quantum

mechanics and cosmology, a recurrent dream of many physicists. Three-dimensional topology was already investigated by several groups in the context of quantum information [6,7], high energy physics [8,9], biology [10] and consciousness studies [11].

Recall the context of our work and clarify its motivation. Bravyi & Kitaev introduced the principle of 'magic state distillation': universal quantum computation, the possibility to implement an arbitrary quantum gate, may be realized thanks to the stabilizer formalism (Clifford group unitaries, preparations and measurements) and the ability to prepare an appropriate single qubit non-stabilizer state, called a 'magic state' [12]. Then, irrespectively of the dimension of the Hilbert space where the quantum states live, a non-stabilizer pure state was called a magic state [13]. An improvement of this concept was carried out in [14,15] showing that a magic state could be at the same time a fiducial state for the construction of an informationally complete positive operator-valued measure, or IC-POVM, under the action on it of the Pauli group of the corresponding dimension. Thus UQC in this view happens to be relevant both to such magic states and to IC-POVMs. In [14,15], a *d*-dimensional magic state follows from the permutation group that organizes the cosets of a subgroup *H* of index *d* of a two-generator free group *G*. This is due to the fact that a permutation may be seen as a permutation matrix/gate and that mutually commuting matrices share eigenstates—they are either of the stabilizer type (as elements of the Pauli group) or of the magic type. In the calculation, it is enough to keep magic states that are simultaneously fiducial states for an IC-POVM and we are done. Remarkably, a rich catalog of the magic states relevant to UQC and IC-POVMs can be obtained by selecting *G* as the two-letter representation of the modular group $\Gamma = PSL(2, \mathbb{Z})$ [16]. The next step, developed in this paper, is to relate the choice of the starting group *G* to three-dimensional topology. More precisely, *G* is taken as the fundamental group $\pi_1(S^3 \setminus K)$ of a 3-manifold $M^3$ defined as the complement of a knot or link *K* in the 3-sphere $S^3$. A branched covering of degree *d* over the selected $M^3$ has a fundamental group corresponding to a subgroup of index *d* of $\pi_1$ and may be identified as a sub-manifold of $M^3$, the one leading to an IC-POVM is a model of UQC. In the specific case of $\Gamma$, the knot involved is the left-handed trefoil knot $T_1$, as shown in Section 2.

While $\Gamma$ serves as a motivation for investigating the trefoil knot manifold in relation to UQC and the corresponding ICs, it is important to put the UQC problem in the wider frame of Poincaré conjecture, the Thurston's geometrization conjecture and the related 3-manifolds [1]. For example, ICs may also follow from hyperbolic or Seifert 3-manifolds as shown in Tables of this paper.

More details are provided at the next subsections.

*1.1. From Poincaré Conjecture to UQC*

The Poincaré conjecture is the elementary (but deep) statement that every simply connected, closed 3-manifold is homeomorphic to the 3-sphere $S^3$ [17]. Having in mind the correspondence between $S^3$ and the Bloch sphere that houses the qubits $\psi = a \left| 0 \right\rangle + b \left| 1 \right\rangle$, $a, b \in \mathbb{C}$, $|a|^2 + |b|^2 = 1$, one would desire a quantum translation of this statement. For doing this, one may use the picture of the Riemann sphere $\mathbb{C} \cup \infty$ in parallel to that of the Bloch sphere and follow F. Klein lectures on the icosahedron to perceive the platonic solids within the landscape [18]. This picture fits well the Hopf fibrations [19], their entanglements described in [20,21] and quasicrystals [22,23]. However, we can be more ambitious and dress $S^3$ in an alternative way that reproduces the historic thread of the proof of Poincaré conjecture. Thurston's geometrization conjecture, from which Poincaré conjecture follows, dresses $S^3$ as a 3-manifold not homeomorphic to $S^3$. The wardrobe of 3-manifolds $M^3$ is huge but almost every dress is hyperbolic and W. Thurston found the recipes for them [1]. Every dress is identified thanks to a signature in terms of invariants. For our purpose, the fundamental group $\pi_1$ of $M^3$ does the job.

The three-dimensional space surrounding a knot *K*—the knot complement $S^3 \setminus K$—is an example of a three-manifold [1,24]. We will be especially interested by the trefoil knot that underlies work of the first author [16] as well as the figure-of-eight knot, the Whitehead link and the Borromean rings because they are universal (in a sense described below), hyperbolic and allow to build 3-manifolds

from platonic manifolds [25]. Such manifolds carry a quantum geometry corresponding to quantum computing and (possibly informationally complete) POVMs identified in our earlier work [14–16].

According to [26], the knot $K$ and the fundamental group $G = \pi_1(S^3 \setminus K)$ are universal if every closed and oriented 3-manifold $M^3$ is homeomorphic to a quotient $\mathbb{H}/G$ of the hyperbolic 3-space $\mathbb{H}$ by a subgroup $H$ of finite index $d$ of $G$. As just announced, the figure-of-eight knot, the Whitehead link and Borromean rings are universal. The catalog of the finite index subgroups of their fundamental group $G$ and of the corresponding 3-manifolds defined from the $d$-fold coverings [27] can easily be established up to degree 8, using the software SnapPy [28].

In paper [16] of the first author, it has been found that minimal $d$-dimensional IC-POVMs (sometimes called MICs) may be built from finite index subgroups of the modular group $\Gamma = PSL(2, \mathbb{Z})$. To such an IC (or MIC) is associated a subgroup of index $d$ of $\Gamma$, a fundamental domain in the Poincaré upper-half plane and a signature in terms of genus, elliptic points and cusps as summarized in ([16] Figure 1). There exists a relationship between the modular group $\Gamma$ and the trefoil knot $T_1$ since the fundamental group $\pi_1(S^3 \setminus T_1)$ of the knot complement is the braid group $B_3$, the central extension of $\Gamma$. However, the trefoil knot and the corresponding braid group $B_3$ are not universal [29] which forbids the relation of the finite index subgroups of $B_3$ to all three-manifolds.

It is known that two coverings of a manifold $M$ with fundamental group $G = \pi_1(M)$ are equivalent if there exists a homeomorphism between them. Besides, a $d$-fold covering is uniquely determined by a subgroup of index $d$ of the group $G$ and the inequivalent $d$-fold coverings of $M$ correspond to conjugacy classes of subgroups of $G$ [27]. In this paper we will fuse the concepts of a three-manifold $M^3$ attached to a subgroup $H$ of index $d$ and the POVM, possibly informationally complete (IC), found from $H$ (thanks to the appropriate magic state and related Pauli group factory).

*1.2. Minimal Informationally Complete POVMs and UQC*

In our approach [15,16], minimal informationally complete (IC) POVMs are derived from appropriate fiducial states under the action of the (generalized) Pauli group. The fiducial states also allow to perform universal quantum computation [14].

A POVM is a collection of positive semi-definite operators $\{E_1, \ldots, E_m\}$ that sum to the identity. In the measurement of a state $\rho$, the $i$-th outcome is obtained with a probability given by the Born rule $p(i) = \mathrm{tr}(\rho E_i)$. For a minimal IC-POVM (or MIC), one needs $d^2$ one-dimensional projectors $\Pi_i = |\psi_i\rangle \langle \psi_i|$, with $\Pi_i = dE_i$, such that the rank of the Gram matrix with elements $\mathrm{tr}(\Pi_i \Pi_j)$, is precisely $d^2$. A SIC-POVM (the $S$ means symmetric) obeys the relation $|\langle \psi_i | \psi_j \rangle|^2 = \mathrm{tr}(\Pi_i \Pi_j) = \frac{d\delta_{ij}+1}{d+1}$, that allows the explicit recovery of the density matrix as in ([30] Equation (29)).

New minimal IC-POVMs (i.e., whose rank of the Gram matrix is $d^2$) and with Hermitian angles $|\langle \psi_i | \psi_j \rangle|_{i \neq j} \in A = \{a_1, \ldots, a_l\}$ have been discovered [16]. A SIC (i.e., a SIC-POVM) is equiangular with $|A| = 1$ and $a_1 = \frac{1}{\sqrt{d+1}}$. The states encountered are considered to live in a cyclotomic field $\mathbb{F} = \mathbb{Q}[\exp(\frac{2i\pi}{n})]$, with $n = \mathrm{GCD}(d, r)$, the greatest common divisor of $d$ and $r$, for some $r$. The Hermitian angle is defined as $|\langle \psi_i | \psi_j \rangle|_{i \neq j} = \|(\psi_i, \psi_j)\|^{\frac{1}{\deg}}$, where $\|.\|$ means the field norm of the pair $(\psi_i, \psi_j)$ in $\mathbb{F}$ and deg is the degree of the extension $\mathbb{F}$ over the rational field $\mathbb{Q}$ [15].

The fiducial states for SIC-POVMs are quite difficult to derive and seem to follow from algebraic number theory [31].

Except for $d = 3$, the IC-POVMs derived from permutation groups are not symmetric and most of them can be recovered thanks to subgroups of index $d$ of the modular group $\Gamma$ ([16] Table 2).The geometry of the qutrit Hesse SIC is shown in Figure 1a. It follows from the action of the qutrit Pauli group on magic/fiducial states of type $(0, 1, \pm 1)$. For $d = 4$, the action of the two-qubit Pauli group on the magic/fiducial state of type $(0, 1, -\omega_6, \omega_6 - 1)$ with $\omega_6 = \exp(\frac{2i\pi}{6})$ results into a minimal IC-POVM whose geometry of triple products of projectors $\Pi_i$ turns out to correspond to the commutation graph of Pauli operators, see Figure 1b and ([16] Figure 2). For $d = 5$,

the geometry of an IC consists of copies of the Petersen graph reproduced in Figure 1c. For $d = 6$, the geometry consists of components looking like Borromean rings (see [16] Figure 2 and Table 1 below).

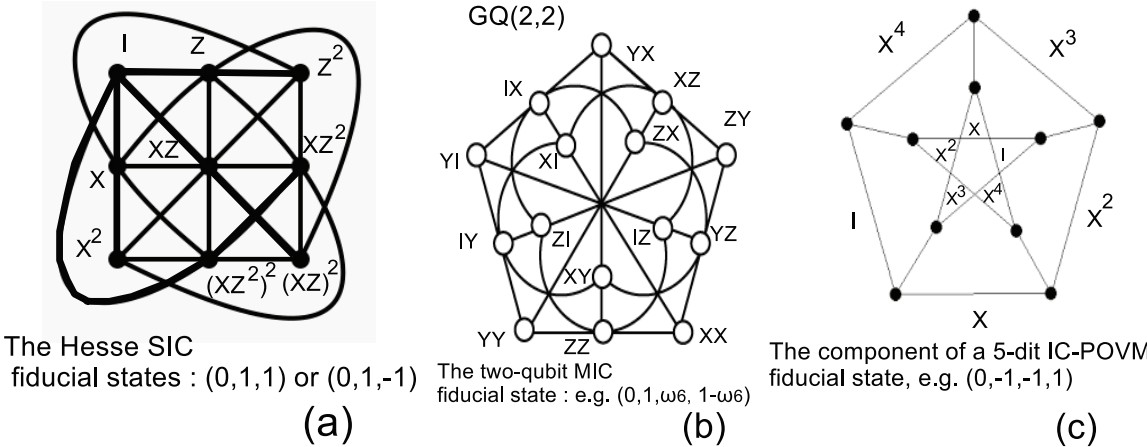

The Hesse SIC
fiducial states : (0,1,1) or (0,1,-1)
**(a)**

The two-qubit MIC
fiducial state : e.g. (0,1,ω6, 1-ω6)
**(b)**

The component of a 5-dit IC-POVM
fiducial state, e.g. (0,-1,-1,1)
**(c)**

**Figure 1.** Geometrical structure of low dimensional MICs: (**a**) the qutrit Hesse SIC, (**b**) the two-qubit MIC that is the generalized quadrangle of order two $GQ(2,2)$, (**c**) the basic component of the 5-dit MIC that is the Petersen graph. The coordinates on each diagram are the $d$-dimensional Pauli operators that act on the fiducial state, as shown.

### 1.3. Organization of the Paper

The paper is organized as follows. Section 2 deals with the relationship between quantum information seen from the modular group $\Gamma$ and from the trefoil knot 3-manifold. Section 3 deals with the (platonic) 3-manifolds related to coverings over the figure-of-eight knot, Whitehead link and Borromean rings, see Figure 2, and how they relate to minimal IC-POVMs. Section 4 describes the important role played by Dehn fillings for describing the many types of 3-manifolds that may relate to topological quantum computing.

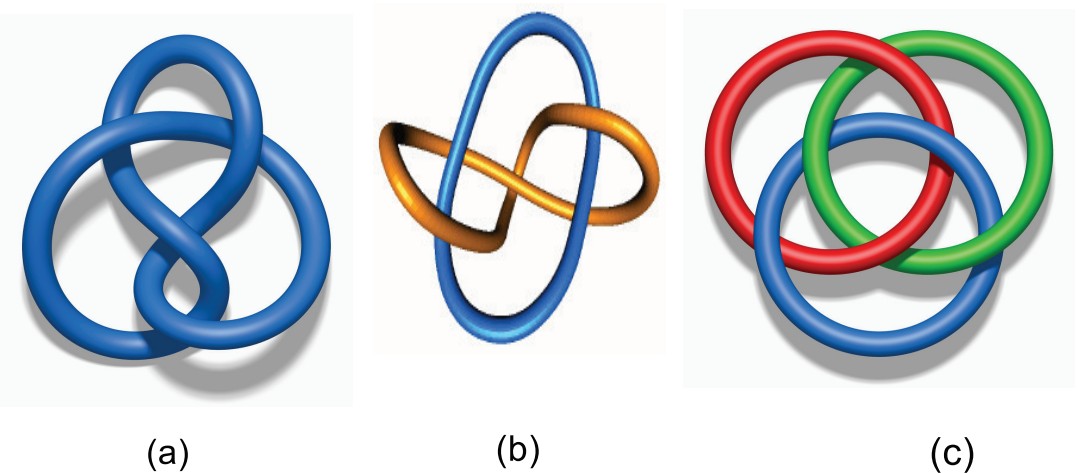

**(a)** **(b)** **(c)**

**Figure 2.** (**a**) The figure-of-eight knot: $K4a1 = otet02_{00001} = m004$, (**b**) the Whitehead link $L5a1 = ooct01_{00001} = m129$, (**c**) Borromean rings $L6a4 = ooct02_{00005} = t12067$.

## 2. Quantum Information from the Modular Group $\Gamma$ and the Related Trefoil Knot $T_1$

In this section, we describe the results established in [16] in terms of the 3-manifolds corresponding to coverings of the trefoil knot complement $S^3 \setminus T_1$.

Let us introduce to the group representation of a knot complement $\pi_1(S^3 \setminus K)$. A Wirtinger representation is a finite representation of $\pi_1$ where the relations are of the form $wg_iw^{-1} = g_j$ where

$w$ is a word in the $k$ generators $\{g_1, \cdots, g_k\}$. For the trefoil knot $T_1 = K3a1 = 3_1$ shown in Figure 3a, a Wirtinger representation is [32]

$$\pi_1(S^3 \setminus T_1) = \langle x, y | yxy = xyx \rangle \quad \text{or equivalently} \quad \pi_1 = \left\langle x, y | y^2 = x^3 \right\rangle.$$

In the rest of the paper, the number of $d$-fold coverings of the manifold $M^3$ corresponding to the knot $T$ will be displayed as the ordered list $\eta_d(T)$, $d \in \{1..10 \ldots\}$. For $T_1$ it is

$$\eta_d(T_1) = \{1, 1, 2, 3, 2, \ 8, 7, 10, 18, 28, \ldots\}.$$

Details about the corresponding $d$-fold coverings are in Table 1. As expected, the coverings correspond to subgroups of index $d$ of the fundamental group associated to the trefoil knot $T_1$.

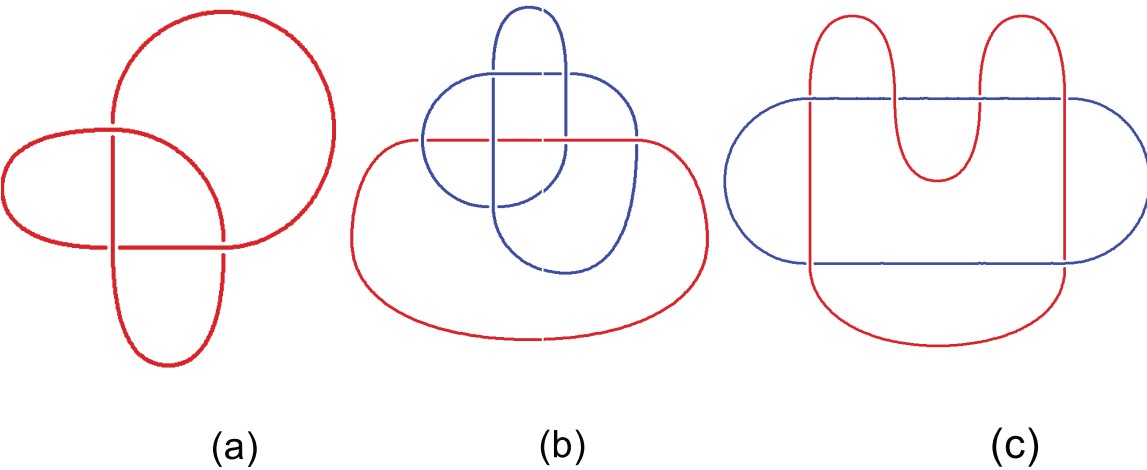

**Figure 3.** (**a**) The trefoil knot $T_1 = K3a1 = 3_1$, (**b**) the link *L7n1* associated to the Hesse SIC, (**c**) the link *L6a3* associated to the two-qubit IC.

### 2.1. Cyclic Branched Coverings over the Trefoil Knot

Let $p, q, r$ be three positive integers (with $p \leq q \leq r$), the Brieskorn 3-manifold $\Sigma(p, q, r)$ is the intersection in $\mathbb{C}^3$ of the 5-sphere $S^5$ with the surface of equation $z_1^p + z_2^q + z_3^r = 1$. In [33], it is shown that a $r$-fold cyclic covering over $S^3$ branched along a torus knot or link of type $(p, q)$ is a Brieskorn 3-manifold $\Sigma(p, q, r)$ (see also Section 4.1). For the spherical case $p^{-1} + q^{-1} + r^{-1} > 1$, the group associated to a Brieskorn manifold is either dihedral [that is the group $D_r$ for the triples $(2, 2, r)$], tetrahedral [that is $A_4$ for $(2, 3, 3)$], octahedral [that is $S_4$ for $(2, 3, 4)$] or icosahedral [that is $A_5$ for $(2, 3, 5)$]. The Euclidean case $p^{-1} + q^{-1} + r^{-1} = 1$ corresponds to $(2, 3, 6)$, $(2, 4, 4)$ or $(3, 3, 3)$. The remaining cases are hyperbolic.

The cyclic branched coverings with spherical groups for the trefoil knot (which is of type $(2, 3)$) are identified in the right hand side column of Table 1.

### 2.2. Irregular branched coverings over the trefoil knot

The right hand side column of Table 1 shows the subgroups of $\Gamma$ identified in ([16] Table 1) as corresponding to a minimal IC-POVM. Let us give a few more details on how to attach a MIC to some coverings/subgroups of the trefoil knot fundamental group $\pi_1(T_1)$. Columns 1 to 6 in Table 1 contain information available in SnapPy [28], with $d$, ty, hom, cp, gens and CS the degree, the type, the first homology group, the number of cusps, the number of generators and the Chern-Simons invariant of the relevant covering, respectively. In column 7, a link is possibly identified by SnapPy when the fundamental group and other invariants attached to the covering correspond to those of the link. For our purpose, we are also interested in the possible recognition of a MIC behind some manifolds in the table.

**Table 1.** Coverings of degree *d* over the trefoil knot found from SnapPy [28]. The related subgroup of modular group Γ and the corresponding IC-POVM [16] (when applicable) is in the right column. The covering is characterized by its type ty, homology group hom (where 1 means $\mathbb{Z}$), the number of cusps cp, the number of generators gens of the fundamental group, the Chern-Simons invariant CS and the type of link it represents (as identified in SnapPy). The links L7n1 (shown in Figure 3b) and L6a3 (shown in Figure 3c) correspond to the Hesse SIC and the two-qubit IC, respectively. The case of cyclic coverings corresponds to Brieskorn 3-manifolds as explained in the text: the spherical groups for these manifolds is given at the right hand side column.

| d | ty | hom | cp | Gens | CS | Link | Type in [16] |
|---|----|-----|----|----|----|----|----|
| 2 | cyc | $\frac{1}{3}+1$ | 1 | 2 | $-1/6$ | | |
| 3 | irr | $1+1$ | 2 | 2 | $1/4$ | L7n1 | $\Gamma_0(2)$, Hesse SIC |
| . | cyc | $\frac{1}{2}+\frac{1}{2}+1$ | 1 | 3 | . | | $A_4$ |
| 4 | irr | $1+1$ | 2 | 2 | $1/6$ | L6a3 | $\Gamma_0(3)$, 2QB IC |
| . | irr | $\frac{1}{2}+1$ | 1 | 3 | . | | $4A^0$, 2QB-IC |
| . | cyc | $\frac{1}{3}+1$ | 1 | 2 | . | | $S_4$ |
| 5 | cyc | $1$ | 1 | 2 | $5/6$ | | $A_5$ |
| . | irr | $\frac{1}{3}+1$ | 1 | 3 | . | | $5A^0$, 5-dit IC |
| 6 | reg | $1+1+1$ | 3 | 3 | $0$ | L8n3 | $\Gamma(2)$, 6-dit IC |
| . | cyc | $1+1+1$ | 1 | 3 | . | | $\Gamma'$, 6-dit IC |
| . | irr | $1+1+1$ | 3 | 3 | . | | |
| . | irr | $\frac{1}{2}+1+1$ | 2 | 3 | . | | $3C^0$, 6-dit IC |
| . | irr | $\frac{1}{2}+1+1$ | 2 | 3 | . | | $\Gamma_0(4)$, 6-dit IC |
| . | irr | $\frac{1}{2}+1+1$ | 2 | 3 | . | | $\Gamma_0(5)$, 6-dit IC |
| . | irr | $\frac{1}{2}+\frac{1}{2}+\frac{1}{2}+1$ | 1 | 4 | . | | |
| . | irr | $\frac{1}{3}+\frac{1}{3}+1$ | 1 | 3 | . | | |
| 7 | cyc | $1$ | 1 | 2 | $-5/6$ | | |
| . | irr | $1+1$ | 2 | 3 | . | | NC 7-dit IC |
| . | irr | $\frac{1}{2}+\frac{1}{2}+1$ | 1 | 4 | . | | $7A^0$ 7-dit IC |
| 8 | irr | $1+1$ | 2 | 2 | $-1/6$ | | |
| . | cyc | $\frac{1}{3}+1$ | 2 | 2 | . | | |
| . | cyc | $\frac{1}{3}+1+1$ | 2 | 3 | . | | |
| . | cyc | $\frac{1}{6}+1$ | 1 | 4 | . | | $8A^0$, $\sim$8-dit IC |

For the irregular covering of degree 3 and first homology $\mathbb{Z}+\mathbb{Z}$, the fundamental group provided by SnapPy is $\pi_1(M^3) = \langle a,b|ab^{-2}a^{-1}b^2\rangle$ that, of course, corresponds to a representative *H* of one of the two conjugacy classes of subgroups of index 3 of the modular group Γ, following the theory of [27]. The organization of cosets of *H* in the two-generator group $G = \langle a,b|a^2,y^3\rangle \cong \Gamma$ thanks to the Coxeter-Todd algorithm (implemented in the software Magma [34]) results in the permutation group $P = \langle 3|(1,2,3),(2,3)\rangle$, as in ([16] Section 3.1). This permutation group is also the one obtained from the congruence subgroup $\Gamma_0(2) \cong S_3$ of Γ (where $S_3$ is the three-letter symmetric group) whose fundamental domain is in ([16] Figure 1b). Then, the eigenstates of the permutation matrix in $S_3$ of type $(0,1,\pm1)$ serve as magic/fiducial state for the Hesse SIC [15,16].

A similar reasoning applied to the irregular coverings of degree 4, and first homology $\mathbb{Z}+\mathbb{Z}$ and $\frac{\mathbb{Z}}{2}+\mathbb{Z}$ leads to the recognition of congruence subgroups $\Gamma_0(3)$ and $4A^0$, respectively, behind the corresponding manifolds. It is known from ([16] Section 3.2) that they allow the construction of two-qubit minimal IC-POVMs. For degree 5, the equiangular 5-dit MIC corresponds to the irregular covering of homology $\frac{\mathbb{Z}}{3}+\mathbb{Z}$ and to the congruence subgroup $5A^0$ in Γ (as in [16] Section 3.3).

Five coverings of degree 6 allow the construction of the (two-valued) 6-dit IC-POVM whose geometry contain the picture of Borromean rings ([16] Figure 2c). The corresponding congruence subgroups of Γ are identified in Table 1. The first, viz $\Gamma(2)$, define a 3-manifold whose fundamental group is the same as the one of the link *L8n3*. The other three coverings leading to the 6-dit IC are the congruence subgroups $\gamma'$, $3C^0$, $\Gamma_0(4)$ and $\Gamma_0(5)$.

## 3. Quantum Information from Universal Knots and Links

In the previous section, we found the opportunity to rewrite results about the existence and construction of $d$-dimensional MICs in terms of the three-manifolds corresponding to some degree $d$ coverings of the trefoil knot $T_1$. However, neither $T_1$ nor the manifolds corresponding to its covering are hyperbolic. In the present section, we proceed with hyperbolic (and universal) knots and links and display the three-manifolds behind the low dimensional MICs. The method is as above in Section 2 in the sense that the fundamental group of a 3-manifold $M^3$ attached to a degree $d$-covering is the one of a representative of the conjugacy class of subgroups of the corresponding index in the relevant knot or link.

### 3.1. Three-Manifolds Pertaining to the Figure-of-Eight Knot

The fundamental group for the figure-of-eight knot $K_0$ is

$$\pi_1(S^3 \setminus K_0) = \left\langle x, y \mid y * x * y^{-1}xy = xyx^{-1}yx \right\rangle.$$

and the number of $d$-fold coverings is in the list

$$\eta_d(K_0) = \{1, 1, 1, 2, 4, \ 11, 9, 10, 11, 38, \ldots\}.$$

Table 2 establishes the list of 3-manifolds corresponding to subgroups of index $d \leq 7$ of the universal group $G = \pi_1(S^3 \setminus K_0)$. The manifolds are labeled otet$N_n$ in [25] because they are oriented and built from $N = 2d$ tetrahedra, with $n$ an index in the table. The identification of 3-manifolds of finite index subgroups of $G$ was first obtained by comparing the cardinality list $\eta_d(H)$ of the corresponding subgroup $H$ to that of a fundamental group of a tetrahedral manifold in SnapPy table [28]. However, there is a more straightforward way to perform this task by identifying a subgroup $H$ to a degree $d$ covering of $K_0$ [27]. The full list of $d$-branched coverings over the figure eight knot up to degree 8 is available in SnapPy. Extra invariants of the corresponding $M^3$ may be found there. In addition, the lattice of branched coverings over $K_0$ was investigated in [35].

**Table 2.** Table of 3-manifolds $M^3$ found from subgroups of finite index $d$ of the fundamental group $\pi_1(S^3 \setminus K_0)$ (alias the $d$-fold coverings of $K_0$). The terminology in column 3 is that of Snappy [28]. The identified $M^3$ is made of $2d$ tetrahedra and has cp cusps. When the rank $rk$ of the POVM Gram matrix is $d^2$ the corresponding IC-POVM shows $pp$ distinct values of pairwise products as shown.

| d | ty | $M^3$ | cp | rk | pp | Comment |
|---|-----|-------|-----|------|-----|---------|
| 2 | cyc | otet04$_{00002}$, $m206$ | 1 | 2 | | |
| 3 | cyc | otet06$_{00003}$, $s961$ | 1 | 3 | | |
| 4 | irr | otet08$_{00002}$, $L10n46$, $t_{12840}$ | 2 | 4 | | Mom-4s [36] |
|   | cyc | otet08$_{00007}$, $t12839$ | 1 | 16 | 1 | 2-qubit IC |
| 5 | cyc | otet10$_{00019}$ | 1 | 21 | | |
|   | irr | otet10$_{00006}$, $L8a20$ | 3 | 15, 21 | | |
|   | irr | otet10$_{00026}$ | 2 | 25 | 1 | 5-dit IC |
| 6 | cyc | otet12$_{00013}$ | 1 | 28 | | |
|   | irr | otet12$_{00041}$ | 2 | 36 | 2 | 6-dit IC |
|   | irr | otet12$_{00039}$, otet12$_{00038}$ | 1 | 31 | | |
|   | irr | otet12$_{00017}$ | 2 | 33 | | |
|   | irr | otet12$_{00000}$ | 2 | 36 | 2 | 6-dit IC |
| 7 | cyc | otet14$_{00019}$ | 1 | 43 | | |
|   | irr | otet14$_{00002}$, $L14n55217$ | 3 | 49 | 2 | 7-dit IC |
|   | irr | otet14$_{00035}$ | 1 | 49 | 2 | 7-dit IC |

Let us give more details about the results summarized in Table 2. Using Magma, the conjugacy class of subgroups of index 2 in the fundamental group $G$ is represented by the subgroup on three generators and two relations as follows $H = \langle x, y, z | y^{-1}zx^{-1}zy^{-1}x^{-2}, z^{-1}yxz^{-1}yz^{-1}xy \rangle$, from which the sequence of subgroups of finite index can be found as $\eta_d(M^3) = \{1, 1, 5, 6, 8, 33, 21, 32, \cdots\}$. The manifold $M^3$ corresponding to this sequence is found in Snappy as otet04$_{00002}$, alias $m206$.

The conjugacy class of subgroups of index 3 in $G$ is represented as

$$H = \left\langle x, y, z | x^{-2}zx^{-1}yz^2x^{-1}zy^{-1}, z^{-1}xz^{-2}xz^{-2}y^{-1}x^{-2}zy \right\rangle,$$

with $\eta_d(M^3) = \{1, 7, 4, 47, 19, 66, 42, 484, \cdots\}$ corresponding to the manifold otet06$_{00003}$, alias $s961$.

As shown in Table 2, there are two conjugacy classes of subgroups of index 4 in $G$ corresponding to tetrahedral manifolds otet08$_{00002}$ (the permutation group $P$ organizing the cosets is $\mathbb{Z}_4$) and otet08$_{00007}$ (the permutation group organizing the cosets is the alternating group $A_4$). The latter group/manifold has fundamental group

$$H = \left\langle x, y, z | yx^{-1}y^{-1}z^{-1}xy^{-2}xyzx^{-1}y, zx^{-1}yx^{-1}yx^{-1}zyx^{-1}y^{-1}z^{-1}xy^{-1} \right\rangle,$$

with cardinality sequences of subgroups as $\eta_d(M^3) = \{1, 3, 8, 25, 36, 229, 435 \cdots\}$. To $H$ is associated an IC-POVM [15,16] which follows from the action of the two-qubit Pauli group on a magic/fiducial state of type $(0, 1, -\omega_6, \omega_6 - 1)$, with $\omega_6 = \exp(2i\pi/6)$ a six-root of unity.

For index 5, there are three types of 3-manifolds corresponding to the subgroups $H$. The tetrahedral manifold otet10$_{00026}$ of cardinality sequence $\eta_d(M^3) = \{1, 7, 15, 88, 123, 802, 1328 \cdots\}$, is associated to a 5-dit equiangular IC-POVM, as in ([15] Table 5).

For index 6, the 11 coverings define six classes of 3-manifolds and two of them: otet12$_{00041}$ and otet12$_{00000}$ are related to the construction of ICs. For index 7, one finds three classes of 3-manifolds with two of them: otet14$_{00002}$ (alias $L14n55217$) and otet14$_{00035}$ are related to ICs. Finally, for index 7, 3 types of 3-manifolds exist, two of them relying on the construction of the 7-dit (two-valued) IC. For index 8, there exists 6 distinct 3-manifolds (not shown) none of them leading to an IC.

A Two-Qubit Tetrahedral Manifold

The tetrahedral three-manifold otet08$_{00007}$ is remarkable in the sense that it corresponds to the subgroup of index 4 of $G$ that allows the construction of the two-qubit IC-POVM. The corresponding hyperbolic polyhedron taken from SnapPy is shown in Figure 4a. Of the 29 orientable tetrahedral manifolds with at most 8 tetrahedra, 20 are two-colorable and each of those has at most 2 cusps. The 4 three-manifolds (with at most 8 tetrahedra) identified in Table 2 belong to the 20's and the two-qubit tetrahedral manifold otet08$_{00007}$ is one with just one cusp ([37] Table 1).

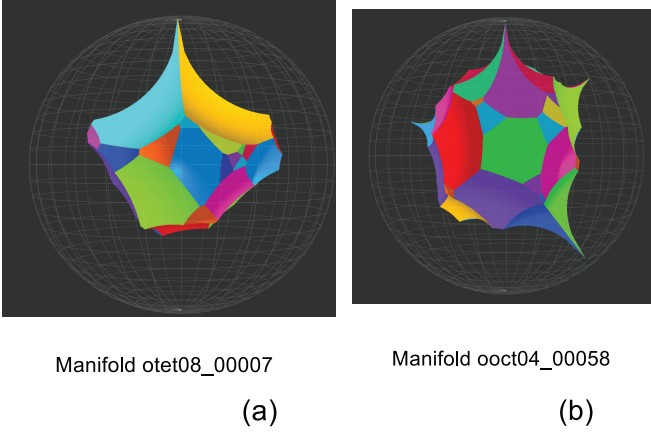

Manifold otet08_00007

Manifold ooct04_00058

(a)        (b)

**Figure 4.** Two platonic three-manifolds leading to the construction of the two-qubit MIC. Details are given in Tables 2 and 3.

**Table 3.** A few 3-manifolds $M^3$ found from subgroups of the fundamental group associated to the Whitehead link. For $d \geq 4$, only the $M^3$'s leading to an IC are listed.

| d | ty | $M^3$ | cp | rk | pp | Comment |
|---|-----|-------|----|----|----|---------|
| 2 | cyc | ooct02$_{00003}$, $t12066$, $L8n5$ | 3 | 2 | | Mom-4s [36] |
|   | cyc | ooct02$_{00018}$, $t12048$ | 2 | 2 | | Mom-4s [36] |
| 3 | cyc | ooct03$_{00011}$, $L10n100$ | 4 | 3 | | |
|   | cyc | ooct03$_{00018}$ | 2 | 3 | | |
|   | irr | ooct03$_{00014}$, $L12n1741$ | 3 | 9 | 1 | qutrit Hesse SIC |
| 4 | irr | ooct04$_{00058}$ | 4 | 16 | 2 | 2-qubit IC |
|   | irr | ooct04$_{00061}$ | 3 | 16 | 2 | 2-qubit IC |
| 5 | irr | ooct05$_{00092}$ | 3 | 25 | 1 | 5-dit IC |
|   | irr | ooct05$_{00285}$ | 2 | 25 | 1 | 5-dit IC |
|   | irr | ooct05$_{00098}$, $L13n11257$ | 4 | 25 | 1 | 5-dit IC |
| 6 | cyc | ooct06$_{06328}$ | 5 | 36 | 2 | 6-dit IC |
|   | irr | ooct06$_{01972}$ | 3 | 36 | 2 | 6-dit IC |
|   | irr | ooct06$_{00471}$ | 4 | 36 | 2 | 6-dit IC |

*3.2. Three-Manifolds Pertaining to the Whitehead Link*

One could also identify the 3-manifold substructure of another universal object, viz the Whitehead link $L_0$ [38].

The cardinality list corresponding to the Whitehead link group $\pi_1(L_0)$ is

$$\eta_d(L_0) = \{1, 3, 6, 17, 22, \ 79, 94, 412, 616, 1659 \ldots\},$$

Table 3 shows that the identified 3-manifolds for index $d$ subgroups of $\pi_1(L_0)$ are aggregates of $d$ octahedra. In particular, one finds that the qutrit Hesse SIC can be built from ooct03$_{00014}$ and that the two-qubit IC-POVM may be built from ooct04$_{00058}$. The hyperbolic polyhedron for the latter octahedral manifold taken from SnapPy is shown in Figure 4b. The former octahedral manifold follows from the link $L12n1741$ shown in Figure 5a and the corresponding polyhedron taken from SnapPy is shown in Figure 5b.

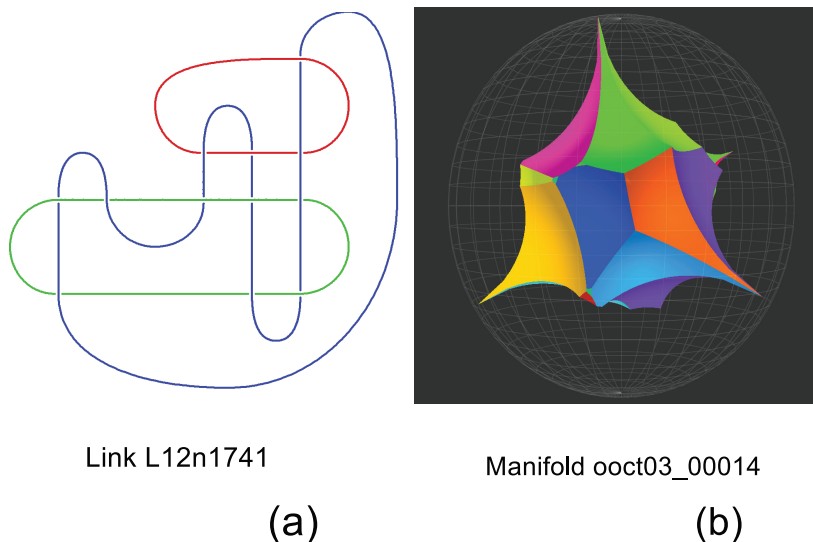

Link L12n1741

Manifold ooct03_00014

(a)                                   (b)

**Figure 5.** (**a**) The link $L12n1741$ associated to the qutrit Hesse SIC, (**b**) The octahedral manifold ooct03$_{00014}$ associated to the 2-qubit IC.

*3.3. A Few Three-Manifolds Pertaining to Borromean Rings*

Three-manifolds corresponding to coverings of degree 2 and 3 of the 3-manifold branched along the Borromean rings *L6a*4 (that is a not a (3,3)-torus link but an hyperbolic link) (see Figure 1c) are given in Table 4. The identified manifolds are hyperbolic octahedral manifolds of volume 14.655 (for the degree 2) and 21.983 (for the degree 3).

**Table 4.** Coverings of degrees 2 to 4 branched over the Borromean rings. The identification of the corresponding hyperbolic 3-manifold $M^3$ is at the 5th column. Only two types of 3-manifolds allow the building of the Hesse SIC. The two 3-manifolds of degree 4 allow the construction of the two-qubit MIC to be identified by the cardinality structure of their subgroups/coverings.

| d | ty | hom | cp | $M^3$ | Comment |
|---|---|---|---|---|---|
| 2 | cyc | $\frac{1}{2} + \frac{1}{2} + 1 + 1 + 1$ | 3 | $ooct04_{00259}$ | |
| . | . | $\frac{1}{2} + 1 + 1 + 1 + 1$ | 4 | $ooct04_{00055}$ | |
| . | . | $1 + 1 + 1 + 1 + 1$ | 5 | $ooct04_{00048}$, $L12n2226$ | |
| 3 | cyc | $\frac{1}{3} + \frac{1}{3} + 1 + 1 + 1$ | 3 | $ooct06_{07427}$ | |
| . | . | $\frac{1}{3} + 1 + 1 + 1 + 1 + 1 + 1$ | 5 | $ooct06_{00463}$ | |
| . | . | $1 + 1 + 1 + 1 + 1 + 1 + 1 + 1$ | 7 | $ooct06_{00411}$ | |
| . | irr | $1 + 1 + 1 + 1$ | 4 | $ooct06_{00466}$ | Hesse SIC |
| . | . | $1 + 1 + 1 + 1 + 1 + 1$ | 4 | $ooct06_{00398}$ | Hesse SIC |
| . | . | $1 + 1 + 1 + 1 + 1 + 1$ | 5 | $ooct06_{00407}$, L14n63856 | |
| 4 | irr | $\frac{1}{2} + \frac{1}{2} + 1 + 1 + 1 + 1$ | 4 | $\{63, 300, 10747 \cdots\}$ | 2QB MIC |
| . | . | $\frac{1}{2} + 1 + 1 + 1 + 1 + 1 + 1$ | 4 | $\{127, 2871, 478956, \cdots\}$ | 2QB MIC |

## 4. A Few Dehn Fillings and Their POVMs

To summarize our findings of the previous section, we started from a building block, a knot (viz the trefoil knot $T_1$) or a link (viz the figure-of-eight knot $K_0$) whose complement in $S^3$ is a 3-manifold $M^3$. Then a *d*-fold covering of $M^3$ was used to build a *d*-dimensional POVM, possibly an IC. Now we apply a kind of 'phase surgery' on the knot or link that transforms $M^3$ and the related coverings while preserving some of the POVMs in a way to be determined. We will start with our friend $T_1$ and arrive at a few standard 3-manifolds of historic importance, the Poincaré homology sphere [alias the Brieskorn sphere $\Sigma(2,3,5)$], the Brieskorn sphere $\Sigma(2,3,7)$ and a Seifert fibered toroidal manifold $\Sigma'$. Then we introduce the 3-manifold $\Sigma_Y$ resulting from 0-surgery on the figure-of-eight knot [39]. Later in this section, we will show how to use the $\{3, 5, 3\}$ Coxeter lattice and surgery to arrive at a hyperbolic 3-manifold $\Sigma_{120e}$ of maximal symmetry whose several coverings (and related POVMs) are close to the ones of the trefoil knot [40].

Let us start with a Lens space $L(p, q)$ that is 3-manifold obtained by gluing the boundaries of two solid tori together, so that the meridian of the first solid torus goes to a $(p, q)$-curve on the second solid torus [where a $(p, q)$-curve wraps around the longitude $p$ times and around the meridian $q$ times]. Then we generalize this concept to a knot exterior, i.e., the complement of an open solid torus knotted like the knot. One glues a solid torus so that its meridian curve goes to a $(p, q)$-curve on the torus boundary of the knot exterior, an operation called Dehn surgery ([1] (p. 275), [24] (p. 259), [41]). According to Lickorish's theorem, every closed, orientable, connected 3-manifold is obtained by performing Dehn surgery on a link in the 3-sphere. For example, surgeries on the trefoil knot allow to build the most important spherical 3-manifolds—the ones with a finite fundamental group—that are the basis of ADE correspondence. The acronym ADE refers to simply laced Dynkin diagrams that connect apparently different objects such as Lie algebras, binary polyhedral groups, Arnold's theory of catastophes, Brieskorn spheres and quasicrystals, to mention a few [42].

### 4.1. A Few Surgeries on the Trefoil Knot

The Poincaré Homology Sphere

The Poincaré dodecahedral space (alias the Poincaré homology sphere) was the first example of a 3-manifold not the 3-sphere. It can be obtained from $(-1,1)$ surgery on the left-handed trefoil knot $T_1$ [43].

Let $p,q,r$ be three positive integers and mutually coprime, the Brieskorn sphere $\Sigma(p,q,r)$ is the intersection in $\mathbb{C}^3$ of the 5-sphere $S^5$ with the surface of equation $z_1^p + z_2^q + z_3^r = 1$. The homology of a Brieskorn sphere is that of the sphere $S^3$. A Brieskorn sphere is homeomorphic but not diffeomorphic to $S^3$. The sphere $\Sigma(2,3,5)$ may be identified to the Poincaré homology sphere. The sphere $\Sigma(2,3,7)$ [39] may be obtained from $(1,1)$ surgery on $T_1$. Table 5 provides the sequences $\eta_d$ for the corresponding surgeries $(\pm 1, 1)$ on $T_1$. Plain digits in these sequences point out the possibility of building ICs of the corresponding degree. This corresponds to a considerable filtering of the ICs coming from $T_1$.

**Table 5.** A few surgeries (column 1), their name (column 2) and the cardinality list of $d$-coverings (alias conjugacy classes of subgroups). Plain characters are used to point out the possible construction of an IC-POVM in at least one the corresponding three-manifolds (see [16] and Section 2 for the ICs corresponding to $T_1$).

| T | Name | $\eta_d(T)$ |
|:---:|:---:|:---:|
| $T_1$ | trefoil | {1,1,**2**,**3**,**2**, **8**,**7**,10,**10**,**28**, **27**,**88**,**134**,**171**,**354**} |
| $T_1(-1,1)$ | $\Sigma(2,3,5)$ | {1,0,0,0,**1**, **1**,0,0,0,**1**, 0,1,0,0,**1**} |
| $T_1(1,1)$ | $\Sigma(2,3,7)$ | {1,0,0,0,0, 0,**2**,1,**1**,0, 0,0,0,9,3} |
| $T_1(0,1)$ | $\Sigma'$ | {1,1,**2**,**2**,1, **5**,**3**,2,**4**,1, 1,12,3,3,**4**} |
| $K_0(0,1)$ | $\Sigma_Y$ | {1,1,1,**2**,2, **5**,1,2,2,**4**, **3**,17,1,1,2} |
| $v_{2413}(-3,2)$ | $\Sigma_{120e}$ | {1,1,**1**,**4**,1, 7,2,25,**3**,**10**, **10**,**62**,1,30,23} |

For instance, the smallest IC from $\Sigma(2,3,5)$ has dimension five and is precisely the one coming from the congruence subgroup $5A^0$ in Table 1. However, it is built from a non modular (fundamental) group whose permutation representation of the cosets is the alternating group $A_5 \cong \langle (1,2,3,4,5), (2,4,3) \rangle$ (compare [15] Section 3.3).

The smallest dimensional IC derived from $\Sigma(2,3,7)$ is 7-dimensional and two-valued, the same as the one arising from the congruence subgroup $7A^0$ given in Table 1. However, it arises from a non modular (fundamental) group with the permutation representation of cosets as $PSL(2,7) \cong \langle (1,2,4,6,7,5,3), (2,5,3)(4,6,7) \rangle$.

### 4.2. The Seifert Fibered Toroidal Manifold $\Sigma'$

An hyperbolic knot (or link) in $S^3$ is one whose complement is 3-manifold $M^3$ endowed with a complete Riemannian metric of constant negative curvature, i.e., it has a hyperbolic geometry and finite volume. A Dehn surgery on a hyperbolic knot is exceptional if it is reducible, toroidal or Seifert fibered (comprising a closed 3-manifold together with a decomposition into a disjoint union of circles called fibers). All other surgeries are hyperbolic. These categories are exclusive for a hyperbolic knot. In contrast, a non-hyperbolic knot such as the trefoil knot admits a toroidal Seifert fiber surgery $\Sigma'$ obtained by $(0,1)$ Dehn filling on $T_1$ [44].

The smallest dimensional ICs built from $\Sigma'$ are the Hesse SIC that is obtained from the congruence subgroup $\Gamma_0(2)$ (as for the trefoil knot) and the two-qubit IC that comes from a non modular fundamental group [with cosets organized as the alternating group $A_4 \cong \langle (2,4,3), (1,2,3) \rangle$].

### 4.3. Akbulut's Manifold $\Sigma_Y$

Exceptional Dehn surgery at slope $(0,1)$ on the figure-of-eight knot $K_0$ leads to a remarkable manifold $\Sigma_Y$ found in [39] in the context of 3-dimensional integral homology spheres smoothly bounding integral homology balls. Apart from its topological importance, we find that some of its coverings are associated to already discovered ICs and those coverings have the same fundamental group $\pi_1(\Sigma_Y)$.

The smallest IC-related covering (of degree 4) occurs with integral homology $\mathbb{Z}$ and the congruence subgroup $\Gamma_0(3)$ also found from the trefoil knot (see Table 1). Next, the covering of degree 6 and homology $\frac{\mathbb{Z}}{5} + \mathbb{Z}$ leads to the 6-dit IC of type $3C^0$ (also found from the trefoil knot). The next case corresponds to the (non-modular) 11-dimensional (3-valued) IC.

### 4.4. The Hyperbolic Manifold $\Sigma_{120e}$

The hyperbolic manifold closest to the trefoil knot manifold known to us was found in [40]. The goal in [40] is the search of—maximally symmetric—fundamental groups of 3-manifolds. In two dimensions, maximal symmetry groups are called Hurwitz groups and arise as quotients of the $(2,3,7)$-triangle groups. In three dimensions, the quotients of the minimal co-volume lattice $\Gamma_{min}$ of hyperbolic isometries, and of its orientation preserving subgroup $\Gamma_{min}^+$, play the role of Hurwitz groups. Let $C$ be the $\{3,5,3\}$ Coxeter group, $\Gamma_{min}$ the split extension $C \rtimes \mathbb{Z}_2$ and $\Gamma_{min}^+$ one of the index two subgroups of $\Gamma_{min}$ of presentation

$$\Gamma_{min}^+ = \left\langle x,y,z | x^3, y^2, z^2, (xyz)^2, (xzyz)^2, (xy)^5 \right\rangle.$$

According to ([40] Corollary 5), all torsion-free subgroups of finite index in $\Gamma_{min}^+$ have index divisible by 60. There are two of them of index 60, called $\Sigma_{60a}$ and $\Sigma_{60b}$, obtained as fundamental groups of surgeries $m017(-4,3)$ and $m016(-4,3)$. Torsion-free subgroups of index 120 in $\Gamma_{min}^+$ are given in Table 6. It is remarkable that these groups are fundamental groups of oriented three-manifolds built with a single icosahedron except for $\Sigma_{120e}$ and $\Sigma_{120g}$.

**Table 6.** The index 120 torsion-free subgroups of $\Gamma_{min}^+$ and their relation to the single isosahedron 3-manifolds [40]. The icosahedral symmetry is broken for $\Sigma_{120e}$ (see the text for details).

| Manifold T | Subgroup | $\eta_d(T)$ |
|---|---|---|
| $oicocld01_{00001} = s897(-3,2)$ | $\Sigma_{120a}$ | $\{1,0,0,0,0,\ 8,2,1,1,8\}$ |
| $oicocld01_{00000} = s900(-3,2)$ | $\Sigma_{120b}$ | $\{1,0,0,0,5,\ 8,10,15,5,24\}$ |
| $oicocld01_{00003} = v2051(-3,2)$ | $\Sigma_{120c}$ | $\{1,0,0,0,0,\ 4,8,12,6,6\}$ |
| $oicocld01_{00002} = s890(3,2)$ | $\Sigma_{120d}$ | $\{1,0,1,5,0,\ 9,0,35,9,2\}$ |
| $v2413(-3,2) \neq oicocld01_{00004}$ | $\Sigma_{120e}$ | $\{1,1,1,4,1,\ 7,2,25,3,10\}$ |
| $oicocld01_{00005} = v3215(1,2)$ | $\Sigma_{120f}$ | $\{1,0,0,0,0,\ 14,10,5,10,17\}$ |
| $v3318(-1,2)$ | $\Sigma_{120g}$ | $\{1,3,1,2,0,\ 11,0,23,12,14\}$ |

The group $\Sigma_{120e}$ is special in the sense that many small dimensional ICs may be built from it in contrast to the other groups in Table 6. The smallest ICs that may be built from $\Sigma_{120e}$ are the Hesse SIC coming from the congruence subgroup $\Gamma_0(2)$, the two-qubit IC coming the congruence subgroup $4A^0$ and the 6-dit ICs coming from the congruence subgroups $\Gamma(2)$, $3C^0$ or $\Gamma_0(4)$ (see [16] Section 3 and Table 1). Higher dimensional ICs found from $\Sigma_{120e}$ do not come from congruence subgroups.

### 5. Conclusions

The relationship between 3-manifolds and universality in quantum computing has been explored in this work. Earlier work of the first author already pointed out the importance of hyperbolic geometry and the modular group $\Gamma$ for deriving the basic small dimensional IC-POVMs. In Section 2, the move from $\Gamma$ to the trefoil knot $T_1$ (and the braid group $B_3$) to non-hyperbolic 3-manifolds could be

investigated by making use of the *d*-fold coverings of $T_1$ that correspond to *d*-dimensional POVMs (some of them being IC). Then, in Section 3, we went on to universal links (such as the figure-of-eight knot, Whitehead link and Borromean rings) and the related hyperbolic platonic manifolds as new models for quantum computing based POVMs. Finally, in Section 4, Dehn fillings on $T_1$ were used to explore the connection of quantum computing to important exotic 3-manifolds (i.e., $\Sigma(2,3,5)$ and $\Sigma(2,3,7)$), to the toroidal Seifert fibered $\Sigma'$, to Akbulut's manifold $\Sigma_Y$ and to a maximum symmetry hyperbolic manifold $\Sigma_{120e}$ slightly breaking the icosahedral symmetry. It is expected that our work will have importance for new ways of implementing quantum computing and for the understanding of the link between quantum information and cosmology [45–47]. A subsequent paper of ours develops the field of 3-manifold based UQC with its relationship to Bianchi groups [48].

**Author Contributions:** All authors contributed significantly to the content of the paper. M.P. wrote the manuscript and the co-authors reviewed it.

**Funding:** The first author acknowledges the support by the French "Investissements d'Avenir" program, project ISITE-BFC (contract ANR-15-IDEX-03). The other resources came from Quantum Gravity Research.

**Conflicts of Interest:** The authors declare no competing interests.

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
