# Peer review of "Universal Quantum Computing and Three-Manifolds"

_symmetry, doi:10.3390/sym10120773_

Round 1

Reviewer 1 Report

The authors have adequately addressed the criticisms in my 

previous report. After the following minor points in the added 

text are also addressed suitably, the paper may be published:

1. P1: The name “William Thurston” is displayed as it were part 

  of the cited text. It should not just be separated by a mere 

  comma, but rather the citation should be closed with a period, 

  and in a new line the name of the author (William Thurston) 

  should be given.

2. P1: “anyon based” should be “anyon-based”

3. P2: “Let us remind the context of our work” should be “... 

  recall ...”.

4. P2: “the possibility to prepare an arbitrary quantum gate” 

  should better be “... to implement ...”, because a quantum gate 

  is an operation rather than a state.

5. P2: On briefly explaining magic states, the author should also 

  take the opportunity to briefly explain “fiducial states”.

6. Figure 2: The font size in the three panel captions should be 

  equal. Also, the third caption starts with “component” and 

  should start with “The component”. Also, “fiducial” should be “

  fiducial state”, and “fiducials” “fiducial states” to avoid 

  jargon.

Author Response

Thank you to reviewer 1 for this complementary comments about our paper. They have been taken into account.

Reviewer 2 Report

The purpose of this paper is to investigate the relationship between 3-manifolds and universality in quantum computing. The topic is current, and the results are correct and significant. The publication of this work is definitely recommended.

Author Response

We are happy that reviewer 2 found our paper significant and correct. Thanks to him.

Reviewer 3 Report

Unfortunately, I can not recommend publishing the current version of this manuscript. The text is so condensed that it is difficult to read. The authors cite a lot of earlier results, but I have doubts if this is necessary. In my opinion, the manuscript could actually begin with section 2. The thesis is not clearly formulated, so it is difficult to consider the argumentation as sufficient. For example, how do  the authors justify the cited link between cosmology and quantum information? However, if the intention of the authors was to write a review article, they also did not achieve the intended goal. In my opinion, the manuscript requires a thorough revision. Above all, the focus should be on clearly presenting the thesis and its justification. Maybe instead of quoting computer-generated tables, one should consider quoting a simple but clearly defined example? I encourage authors to correct the manuscript.

Author Response

Thanks to reviewer 3 for reading the paper. We mention that the original version did not include such a long introduction but it was asked by other previous referees to do the paper more selfcontained by adding such an introduction to the topics developed before (mainly by the first author). Referee 3 is right saying that the content in section 2 is new. Our paper is in no way a review paper. Concerning the calculations that are provided in the tables, they can easily  be reproduced by any reader by following the cecipes provided in the text.

Incidently, we briefly mentioned thet 3-manifolds have found applications in the field of cosmology before, It is not the goal of the paper to address this topic.

Round 2

Reviewer 3 Report

I recommend publication of the revised manuscript.